# The Combined Effects of Sports Smart Bracelet and Multi-Component Exercise Program on Exercise Motivation among the Elderly in Macau

**DOI:** 10.3390/medicina57010034

**Published:** 2021-01-02

**Authors:** Cheuk Kei Lao, Bing Long Wang, Richard S. Wang, Hsiao Yun Chang

**Affiliations:** 1Doctorate Degree of Public Health, Faculty of Medicine, Macau University of Science Technology, Macau 999078, China; joelao818@gmail.com; 2Physiotherapist, Principal, Department of Physiotherapy Rehabilitation, Centro Hospitalar Conde de São Januário, Macau Health Bureau, Macau 999078, China; 3School of Public Health, Peking Union Medical College, Beijing 100730, China; 4School of Healthcare Management, University of Sanya, Sanya 572022, China; 5Affiliation Program of Data Analytics and Business Computing, Stern School of Business, New York University, New York, NY 10012, USA; r851126@gmail.com; 6Department of Athletic Training and Healthy, National Taiwan Sport University, Taoyuan 333019, Taiwan; yun1130@ntsu.edu.tw

**Keywords:** sports smart bracelet, multi-component exercise program, exercise motivation, elderly, Macau

## Abstract

*Background and objectives:* Faced with the serious problem of an aging population, exercise is one of the most effective ways to maintain the health of the elderly. In recent years, with the popularization of smartphones, the elderly have increasingly accepted technological products that incorporate artificial intelligence (AI). However, there is not much research on using artificial intelligence bracelets to enhance elders’ motivation and participation in exercise. Therefore, the purpose of this study is to evaluate the effectiveness of the combination of sports smart bracelets and multi-sport training programs on the motivation of the elderly in Macau. *Materials and Methods:* The study was conducted with a randomized trial design in a 12 week multi-sport exercise training intervention. According to the evaluation, a total of sixty elders’ pre- and post-test data were included in this study. *Results:* After 12 weeks of multi-sport exercise training, the evaluation scores on the exercise motivation scale (EMS) increased significantly in the group wearing exercise bracelets and those taking part in the multi-component exercise program, and the degree of progress reached a statistically significant level, but the control group did not show any statistically significant difference. The influence of the combination of sports smart bracelets and multi-sport training programs on elders’ motivation is clearer. *Conclusions:* The use of sports smart bracelets by elderly people in conjunction with diverse exercise training can effectively enhance elders’ motivation and increase their participation in regular exercise. The combination of sports smart bracelets and multi-sport training programs is worth promoting in the elderly population.

## 1. Introduction

Population aging is a pervasive global problem. According to the United Nations (UN), by the end of 2018 there were approximately 705 million people over the age of 65 in the world [1]. Macau has a very high score in the human development index (HDI) and has the fourth-highest life expectancy in the world [2]. With the increase in age, the physical functions of the body decline [3], including muscle strength, muscle endurance, joint mobility, balance, agility, etc., which leads to the elderly’s inability to take care of themselves in their daily lives. Therefore, they may need medical care, long-term care, and personal assistance from caregivers [4,5]. The literature suggests that regular exercise has a positive effect on delaying the aging process and helping to prevent chronic diseases, reduce the risk of falls, improve quality of life, and promote the physical and mental health of the elderly [6,7]. Thus, participation in sports and proper physical activity are important parts of healthy aging [8]. The types of sports that are beneficial to the elderly should be diversified, including aerobic, resistance, flexibility, and balance training [9,10,11,12].

In recent years, with the popularization and acceptance of artificial intelligence (AI) and other technological products by the elderly, several studies have shown that AI sports devices with exercise programs may improve motivation and related outcomes [13,14]. Macau is transforming into a smart city, and smart bracelets are commonly used to monitor the physiological data generated from body movement; users can have a better understanding of their physical condition during the exercise process. Kononova and coworkers monitored physical activity by using wearable activity trackers for the elderly [15]. The results suggested that wearable activity trackers may improve physical activity levels and internal motivation. They also encouraged older adults to participate in the exercise and compete with other users [15]. Rupp et al. examined the impact of several user characteristics (i.e., personality, age, computer self-efficacy, physical activity level) and device characteristics (trust, usability, and motivational affordances) on the behavioral intentions to use a wearable fitness device. They found computer self-efficacy and physical activity level, as well as personality traits, indirectly increased the desire to use a fitness device and the saliency of perceived motivational affordances [16]. Kononova et al. suggested that activity trackers may be an effective technology for encouraging physical activity among older adults, especially those who have never tried it [15]. Maintenance depended on recognizing the long-term benefits of tracker use, social support, and internal motivation. Non-adoption and relapse may occur because of technology’s limitations and gaining awareness of one’s physical activity without changing the physical activity level itself [15]. Swartz et al. studied older adults with and without involvement in a 12 week tracker-based activity intervention designed to identify related factors associated with weekly adherence. Achieving step goals and receiving virtual support were the important factors that contributed to improved weekly adherence. The authors have suggested the tracker-based activity intervention may promote active lifestyles [17]. Artificial intelligence sports bracelets are smart wearable devices commonly used in exercises today [18]. By collecting data and browsing health records, they can monitor physiological data during physical exercise, and users can have a better understanding of their physical condition during exercise [19].

However, limited empirical research has been conducted using artificial intelligence bracelets to promote elders’ motivation to and participation in exercise. The effects on elderly people of wearing sports smart bracelets are unknown. Thus, this study is the first time that Macau has explored the effects of the combined use of sports smart bracelets by the elderly on the motivation to exercise. According to the innovation and motivation theory [19], the hypothesis was that the use of smart bracelets by the elders in conjunction with diverse exercise training can effectively enhance the elders’ motivation in the community and increase their participation. Therefore, the purpose of this study is to study the effectiveness of the combined use of sports smart bracelets and multi-sports training programs on the motivation for the elderly in the Macau community. This study also provides the Macau Government with recommendations for its future promotion of the Smart and Elderly Campaign to formulate policies and implement programs for the elderly regarding health, achieving the realization of the Macau government’s vision of active aging.

## 2. Methods

### 2.1. Study Design and Sample

The study involved elderly people over 65 years of age in Macau in a randomized controlled trial (RCT) study of the experimental group and control group in a before–after test design. A total of 75 elderly people who received services from the Elderly Centre in Macau were recruited as the study subjects, regardless of gender. Anyone could participate in the study as long as they met the inclusion and exclusion criteria. The inclusion criteria included: no major diseases, normal physical activity, participation in sports, access to all relevant tests for the study, and willingness to participate in sports. In addition, those who were interested in participating in the study, willing to continue to participate in the study, and had no major plans or work arrangements (e.g., immigration, exoduses, surgery, etc.) during the trial period were included. However, those who may have terminated their participation in the study program at the time of recruitment were excluded, reducing the total proportion of sample loss. The intervention period of the elderly’s pluralistic movement was 12 weeks, during which other sports planners could not participate in order to avoid affecting the effectiveness of the intervention of the campaign in this study. The exclusion criteria were as follows: (1) subjects who suffered from problems such as severe heart disease, severe hypertension, etc., which may have made exercise difficult; (2) subjects suffering from any neuroskeletal muscle problems in the limbs, such as amputations and other physical disorders, which may have resulted in the inability to complete the test or to engage in the exercise performed,; and (3) subjects with any neurological disease resulting in the inability to complete the test or to engage in the exercises to be carried out, such as stroke, spinal cord injury, etc.

An appropriate sample size was estimated using power analysis to control for Type II errors. This study used Statistical Software Sample Power 2.0, with the power set at 0.8 to limit the risk of a Type II error to 20%, the Alpha value was set at 0.05, and the covariate’s *R*^2^ was set at 0.13. This was based on the results of one study found in the literature; after a 12 week targeted, multiple intervention, researchers found a mean difference in the exercise motivation scale (EMS) between the two experimental and control groups of 2.0, with a pooled standard deviation of 3.6 [20]. Therefore, the effect size of the covariate adjustment in this study was set at 0.37. Sixty subjects were randomly divided into the exercising while wearing exercise bracelets group, the exercising only group, and the control group, with 20 people in each group. The experimental period was 12 weeks. The group that exercised while wearing exercise bracelets and group that did exercise only were involved in elder diversity training. One group exercised while wearing sports bracelets during the intervention, while another group exercised without using sports bracelets; the control group did not receive any intervention. Before and after the experimental period of 12 weeks, the three groups were compared using the pre-test evaluation of the exercise motivation scale (EMS) scores [21], and relevant data of the pre- and post-assessment tests were systematically analyzed and evaluated. In the study, only the participants who completed the pre-test and post-test of the exercise motivation scale were selected, and the participants in the exercising while wearing exercise bracelets group and the exercising only group required 12 weeks of exercise intervention. The attendance rate reached more than 80%, which met the above conditions as the object of the study.

We undertook the following preparation before the start of the experiment: before the start of the experiment, the first 75 elderly participants in the study were randomly divided into 3 groups of 25 people each. After collecting basic personal information, including age and gender, each elderly person voluntarily signed an informed consent form for the study. The researchers (five registered physiotherapists and five activity coordinators in Macau) were trained in functional physical ability testing, and each experimental group was tested by the same group of researchers in order to complete the elderly motor motivation scale and collect basic data and pre-test scores. To prevent interference with the exercise intervention effect of this study, participants were required not to participate in similar sports training and intervention activities during the study period. The experiment intervention phase involved wearing sports smart bracelets during the 12 weeks of prescription exercise, which involved exercising three times a week for one hour at a time (including warm-up and strain relaxation). The exercise prescription was designed according to the U.S. Department of Health and Human Services [9] and the British Department of Health [10] exercise prescription recommendations (Schedule 3-3-5). The exercise only group (25 people) were involved in 12 weeks of elderly prescription exercise, which was the same as for the group exercising while wearing the bracelet. The control group (25 people) involved in the experiment for 12 weeks did not do any exercise, nor were they equipped with any sort of sports smart bracelets and products.

### 2.2. Research Tools

#### 2.2.1. Assessment Tool

This study used the Exercise Motivation Scale (EMS) [21]. Assessing the overall reliability shows that the EMS has four related subscales relating to the motivation of the elderly to participate in sports. They are: (1) fitness/health management, with a Cronbach’s Alpha level of 0.78, (2) appearance/weight management, with a level of 0.84, (3) stress/emotional management, and (4) social/leisure, with a measure of 0.84 in four components, or, in short: the four degrees of confidence range was Cronbach Alpha 0.78. The full-measurement scale had Cronbach’s Alpha 0.79. The pre-confidence measurement used in this study was Cronbach’s Alpha of 0.955, and the post-confidence measurement was Cronbach’s Alpha of 0.852. The study used the senior motor motivation scale with two alpha confidence coefficients of 0.955 and 0.852, showing good reliability in the elderly’s motor motivation scale. Because the four subscales have a high correlation coefficient range from 0.82–0.93, this study is focused on the overall motivation scale change that occurred after wearing sports bracelets, and we do not analyze the experimental effects on each subscale. In the screening of exercise intensity, the study used the Borg Ratings of Perceived Exertion Scale (RPE scale) rating the continuous intensity of exercise on a scale of 1 to 10. Higher Borg ratings indicate higher intensity [22]. Participants were required to reach at least level 5–6 on the scale or more; that is, they had to at least to reach the “a little hard” moderate exercise intensity [23]. During the study period, participants had to maintain their original living habits in addition to maintaining their original exercise habits and not participating in moderate-intensity or above sports training activities.

#### 2.2.2. Intervention Tool

The sports smart bracelet used in this study was the Mi 3 bracelet, and the 12 week multi-sport exercise for the elderly was based on the recommendations of the US Department of Health and Human Services [9] and the UK Department of Health [10]. The exercise prescription included cardiopulmonary endurance training (25%), muscle strength training (25%), flexibility training (25%), and balance training (25%). Exercise training was carried out 3 times a week for a duration of 60 min each time. Members of the group who wore smart bracelets while exercising were equipped with Mi 3 bracelets. Before the start of the experiment, the functions of the Mi 3 bracelet (heartbeat monitoring, step counting) were introduced to ensure that the team members had a basic understanding of the ring’s components and simple usage techniques. Daycare center staff regularly charged the elderly’s bracelets while downloading bracelet-related data to ensure that the participants were wearing the bracelets throughout the duration of the experiment. The Institute used the Mi 3 smart bracelet. The Mi 3 bracelet was used because it is small, lightweight, cheap ($26), waterproof, durable, easy to wear and use, and convenient to charge. The smart bracelet’s motion sensors automatically detect the user’s activity status and prompt the user to get up when he or she is sedentary. The number of heartbeats was monitored after various activities, which helped us to record and track the frequency of each user’s heartbeat and temperature changes. The OLED display screen on the bracelet shows physical activity data such as current time, number of steps, heart rate, calorie consumption, and more by taping the circular button. The Mi 3 bracelet itself cannot analyze, process, and store large amounts of data. By connecting to Mi 3’s mobile app through the network or Bluetooth, the following data can be learned: 1. Walking distance: records the number or distance of walks per day, which can be converted into calories burned; 2. Exercise Time: records the time spent exercising per day, which can also produce sedentary or immobile time data; 3. Collect sleep status: Records daily sleep status and assesses body state; 4. Calories Burned: calculates calories burned based on the amount of exercise each day; it alerts you when targeted calories burned is not met.

### 2.3. Statistical Analysis

For descriptive statistics, we analyzed variables including age, height, weight, and percentage of study participants in each group with the arithmetic mean and standard deviation of descriptive statistics. We conducted ANOVA and the paired t-test to compare the changes in the scores on the Exercise Motivation Scale before and after each group of experiments. All statistical tests were two-sided with the alpha level set at 0.05. Analyses were done with SPSS, version 22.

## 3. Results

The data of 60 subjects were collected, and the factors of subject withdrawal included physical discomfort, absence due to illness, death, failure to complete 12 weeks of training, failure to complete pre and post-tests, reluctance to wear the sports smart bracelets and other factors (Figure 1).

The background data of the three groups of subjects was similar in age, height, and before and after weight, and there was no significant difference (*p* > 0.05); in terms of gender ratio, the three groups were the same with 1 male and 19 females, and the male to female ratio was the same with no significant difference (Table 1).

The Cronbach’s Alpha reliability coefficients of the Elderly Exercise Motivation Scale were 0.955 > 0.8 and 0.852 > 0.8, indicating that the Elderly Exercise Motivation Scale has good overall reliability. According to whether there was any change on the Elderly Motivation Scale before and after the experiment, an analysis was conducted to find the average of the scale before and after the experiment for each group, and the paired sample t-test was performed.

Table 2 shows the influence of exercising while wearing the bracelet, exercising without a bracelet, and being in the control group on the elders’ motivation of exercise. Before the experiment there was no statistically significant difference between the three groups (*p* > 0.05). For the group who exercised while wearing sports bracelets, there was a significant change in the exercise motivation scale. The average value before the experiment was 3.556, and the average value after the experiment was 4.703. After the 12 week intervention in the group who exercised without bracelets, there was a statistically significant change in the exercise motivation scale of the elderly; the average value before the experiment was 3.146, and the average value after the experiment was 4.165. The post-average value is significantly greater than the pre-test average value, and the level of progression is significantly different (*p* < 0.05).

Before and after the 12th week, the average value of the control group subjects was 4.090 before the experiment and 4.053 after the experiment. The mean value table of the exercise motivation of the elderly before and after the experiment did not produce a significant change, and the change did not reach a statistically significant level (*p* > 0.05).

The analysis of variance (ANOVA) was also used to compare the average values of the group who exercised while wearing the bracelet, the group who exercised without bracelets, and the control group before and after the experiment, as shown in Table 3. There are significant differences between the three groups (*p* < 0.001). The average value of the group who exercised while wearing the bracelet is slightly higher than the average value for the group who just exercised.

The study found that the elderly are concerned about their health and suggested that they should be encouraged to participate in interventional programs of sports courses. Before the start of the exercise training course for the elderly, they should first understand the positive effects and benefits of regular exercise on the body. This would have a positive effect on the recruitment of subjects, on the interventional research program, and on the implementation of the exercise program for the elderly.

## 4. Discussion

After 12 weeks of multi-sport exercise training, the evaluation scores on the elderly’s exercise motivation scale all increased significantly, and the degree of progress reached a significant difference (*p* < 0.05); the control group did not produce significant changes, and the degree did not show a significant difference (*p* > 0.05). From this, it can be inferred that being involved in 12 weeks of multi-sport exercise training can significantly enhance the elderly’s motivation to exercise.

A systematic analysis was conducted based on the comparison of the average before and after the experiment performed on the group who exercised while wearing the bracelet and those who just exercised. The average of the two groups before and after the experiment was significantly different (*p* < 0.05), while the average value of the bracelet-wearing group is slightly higher than that of non-bracelet-wearing group. It can be inferred that in the group who wore bracelets had a significant improvement over the group who just exercised. Wearing sports smart bracelets can significantly improve the elderly’s motivation to exercise.

Motivation is a very important psychological factor in exercise participation. Motivation can be the driving force for elders to participate in fitness activities. It is pointed out that the factors that determine the motivation to participate in exercise include physical, psychological, and social considerations. Physiological factors include height, weight, and health; psychological aspects include self-esteem, emotions and fun, and social aspects include competition and environment.

The participation of elders in fitness exercises can enrich life in their later years. Exercise brings happiness, relaxation, excitement, and other feelings to elders [17]. It is an important source of motivation. Exercises provide elders with interpersonal relationships and friendships, and expand their lives. At the same time, exercises can cultivate the habit of exercise, shift the focus of life, and help elders gain a sense of accomplishment. It helps them with overcoming loneliness, releasing stress, enhancing positive emotions, and overcoming negative emotions such as depression. It also helps the elderly maintain physical fitness and slow down the rate of degradation and disease. The promotion of exercise participation motivation encourages the elderly to participate in exercises, including internal stimuli such as physical fitness and health and external stimuli such as physical performance, psychological emotions, and social interaction [15].

There are several limitations to this research: this research is based on elderly people over 65 years of age, and the ages of the participants ranged from 65 to 95 years of age. There is a large age gap among the elderly, and their physical conditions, functional statuses, and functional levels are very different. Future research should be subdivided into several groups based on the age of the elderly to avoid variation. It is also of note that due to the constraints of time, funding, and human resources, the study was only able to conduct a 12 week analysis of the exercise plan and could not follow more subjects for longer periods. This study adopts the method of free recruitment and random grouping, so it is difficult to control the homogeneity of the sample grouping. In the future, when similar studies are conducted, the group’s homogeneity should be controlled. Future research directions can explore the retention effect of exercises, such as the differences among different exercise intervention methods and the effects of different sports training. The sports smart bracelet records personalized exercise status, and the accumulated data recording daily exercise volume are also worthy of further study. Future smart wearable device technology products will be designed from the perspective of the elderly, which is likewise worthy of further discussion. The research meets the actual use and market needs of the elderly. One of the major limitations was that the available data were unable to perform detailed analysis to tease out the dose–response relationship of the experiment. Therefore, the future study should explore the dose–response relationship between the intensity of AI sports smart bracelets usage and motivational outcomes. Lastly, no objective measures of exercise motivation are available from the study subjects. Future AI research could also benefit from gathering detailed and validated information on compliance with prescribed exercises.

## 5. Conclusions

Exercise has a positive impact on the physical, psychological and social development of the elderly. This study confirms that the elderly can further improve their activity and aging patterns by using smart sports bracelets and a 12 week exercise program to enhance their motivation to exercise, thereby enhancing their level of concern for their health. It was also shown that a 12 week, 60-min exercise program enhanced the motivation of the elderly to exercise in order to achieve an active lifestyle and slow down aging. Therefore, we see that the use of training programs has a positive impact on the promotion of movement for the elderly in Macao, regardless of which training program is used. Wearing sports smart bracelets had an even more significant effect on the elders’ motivation to exercise than exercise alone had.

## Figures and Tables

**Figure 1 medicina-57-00034-f001:**
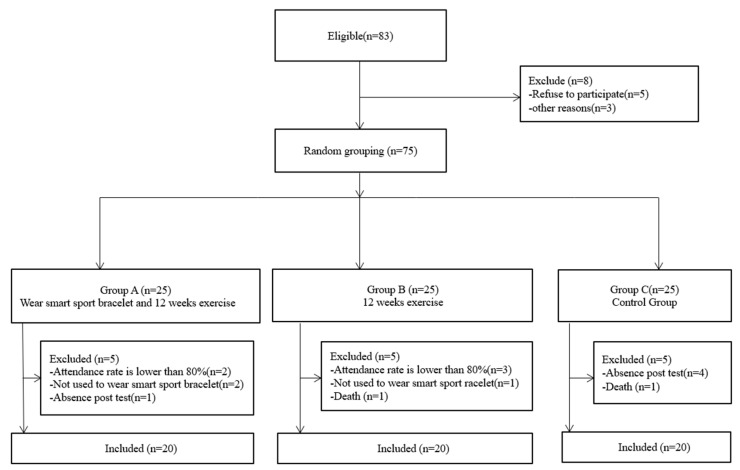
The flowchart of the study of sample selection from a multi-component exercise program among the elderly in Macau.

**Table 1 medicina-57-00034-t001:** Characteristics of the study at baseline.

	Exercising Wearing the Bracelet	Exercising	Control	F	*p*-Value
	Mean (*n* = 20)	Mean (*n* = 20)	Mean (*n* = 20)		
Age (years) Age range(y/o)	80.00 ± 5.38 67–90	78.40 ± 6.75 68–97	76.75 ± 10.64 68–92	0.845	0.435
Heights (cm)	149 ± 0.06	152 ± 0.09	150 ± 0.05	1.274	0.287
Pre-Weight(kg)	55.68 ± 10.05	58.36 ± 11.08	56.55 ± 6.89	0.413	0.664
Post-Weight(kg)	55.33 ± 9.86	56.92 ± 11.23	56.90 ± 6.87	2.705	0.075
Gender	20	20	20		
Male	1	1	1		
Female	19	19	19		

**Table 2 medicina-57-00034-t002:** The differences between the combined use of sports smart bracelets and multi-exercise training programs on the motivation of the elderly.

Group	Intervention	Mean	SD	*t* Value	F	*p*-Value
All	pre-test				2.867	*p* > 0.069
exercising wearing the bracelet	pre-test	3.556	0.623	−8.257		*p* < 0.001
	post-test	4.703	0.085			
exercising	pre-test	3.146	0.574	−7.978		*p* < 0.001 *
	post-test	4.165	0.146			
Control	pre-test	4.090	0.629	0.589		0.563
	post-test	4.053	0.470			

* significantly different at *p* < 0.001.

**Table 3 medicina-57-00034-t003:** Differences between the sports bracelet exercise group and exercise only group.

	SS	Df	MS	F	*p*-Value
SSB	2.889	1	2.889	200.319	*p* < 0.001
SSW	0.548	38	0.014		
SST	3.437	39			

SS: Sum of Squares; MS: Mean Square; SSB: Between-group SUM of Squares Error; SSW: Within-group SUM of Squares Error; SST: Total SUM of Squares Error.

## Data Availability

The study followed MDPI research data policies.

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
