# Peer review of "The Combined Effects of Sports Smart Bracelet and Multi-Component Exercise Program on Exercise Motivation among the Elderly in Macau"

_medicina, 2021, doi:10.3390/medicina57010034_

Round 1

Reviewer 1 Report

I think that all those involved in the sports phenomenon are interested in all the information that adds something to their activity. Sports training lives on creating exercises and motivation on a daily by daily,  since this investigation does not include anything new, it tells us that in older people, innovation and new technologies can be a source of motivation for the exercise.
I think it is something that sports agents need to know.

Reviewer 2 Report

This is a small work regarding the effect of a wearable device in the motivation of elderly in performing exercise. Regarding the EMS scale, it is not clear what is recorded and if there are any subscale differences. Baseline data are peculiar.

Regarding the “intervention” of the bracelet, there is no information. We don’t know about the meaning and the significance of the device in the daily activities of the elderly. We also don’t learn anything about any compliance to the purpose of the device.

Per se, it appears that there is no objective measure of the motivation, as there is no difference in compliance to exercise, or any other physical activity measure. It is not clear it the authors had performed such a screening, but it came back empty.

There are some minor grammar/syntax errors, e.g. combine with, the use of “the” etc. When a sentence starts with a number, please spell the number. There is also no uniform verb tense.

There are additional problems with data analysis and interpretation. There is a need to revise carefully the manuscript and update many fields.

Specific comments

In the abstract, it would be better to remove the reference to SPSS. It is essential to provide age range and sex, the actual data (results), the type of comparison, and the statistical output. It is also essential to provide specifics about the bracelets, that is what type of data are available to the user, how often, and if there is any kind of interaction or scheduled events.

L54 softness =flexibility?

L56-79 It would be preferable to add some details about the reports and the duration of each intervention, and then split the paragraph to discuss compliance.

Details about the type of wearable feedback should be added.

L89 The study involved

L89 Please refer to the RCT protocol

2.1 Please provide participant characteristics, screening, health status.

Did participants consent to this study?

2.2.1 There is a need to rephrase all tools, first the name, then what is measured, then how it is measured, then the outcome and then grading. If the tool has been validated, please provide an extra sentence referring to the characteristics of the study group, Cronbach’s alpha etc.

2.2.2 Separate sentences for the wearable device and the intervention.

First the intervention. Where, when (time of the day), group exercise or not, supervised or not, frequency, duration, details (25% is not appropriate). Add the details about Borg scale, who and how the evaluation was performed, what were the results. Considering that this is a work deriving form a Rehabilitation Department, it is expected to include major details about exercise programs, and prior compliance.

Give specifics about Mi  3  generation  bracelet, company, input, output, interactions.

Should participants wear the device all the time?

Please add details about previous use in any age group, and if possible in the elderly.

L143 remove successfully – There were 20 or 60 participants?

Table 1: Is there any reason that Group B dropped 2 kg?

Did you check your data for normality?

Please use 0.XXX, not .XXX. We also do not need the whole output of SPSS, just a valuable summary.

L154 It is not clear “reliabilities” of what measure have been explored.

Why values in Tables 2& 3 are different?

One Table is enough. Have them combined, and start presenting baseline.

Possibly, values at baseline may be “statistically similar”, but the shift is moving towards control baseline, and this is not very satisfying, as per the randomization process.

Comment in 190-191 understandable, but it should be statistically supported.

I would recommend to present first these data, and then have each score personalized, by substracting post- from pre- values. A paired T test, and testing for the variance of differences, rather than group means, would be preferable.

Occasionally, you may refer to group characteristics: control, exercising, exercising wearing the bracelet, instead of A and B.

L239 Rephrase and extend
